# Comparison of β-1-3-D-Glucan and *Candida* Mannan Biomarker Assays with Serological Tests for the Diagnosis of Candidemia

**DOI:** 10.3390/jof9080813

**Published:** 2023-07-31

**Authors:** Christopher P. Eades, Ahmed Rafezzan Bin Ahmed Bakri, Jeffrey C. Y. Lau, Caroline B. Moore, Lilyann Novak-Frazer, Malcolm D. Richardson, Riina Rautemaa-Richardson

**Affiliations:** 1Department of Infectious Diseases, Manchester University NHS Foundation Trust, Wythenshawe Hospital, Manchester M23 9LT, UK; riina.richardson@manchester.ac.uk; 2Division of Infection, Immunity and Respiratory Medicine, School of Biological Sciences, NIHR Manchester Biomedical Research Centre (BRC) at the Manchester Academic Health Science Centre, The University of Manchester and Manchester University NHS Foundation Trust, Wythenshawe Hospital, Manchester M23 9LT, UKjeffrey.lau@manchester.ac.uk (J.C.Y.L.); lily.novak-frazer@mft.nhs.uk (L.N.-F.); malcolm.richardson@manchester.ac.uk (M.D.R.); 3Mycology Reference Centre Manchester (MRCM), ECMM Excellence Centre of Medical Mycology, Manchester University NHS Foundation Trust, Wythenshawe Hospital, Manchester M23 9LT, UK; 4Manchester Fungal Infection Group (MFIG), Faculty of Biology, Medicine & Health, The University of Manchester, Manchester M13 9NT, UK

**Keywords:** invasive candidiasis, laboratory diagnostics, assay, serology, immunoglobulin, antigen

## Abstract

Invasive candidiasis, including bloodstream infection (candidemia), encompasses the most severe forms of *Candida* infection. Several species-specific and non-specific serological assays are commercially available to aid in diagnosis. This study compared the performance of five such biomarker assays. Serum samples from 14 patients with proven or probable invasive candidiasis, and from 10 control patients, were included in the analysis. A total of 50 serum samples were tested using *C. albicans* germ tube antibody (CAGTA) assay (Vircell), *C. albicans* IgM, *C. albicans* IgG and *Candida* mannan assays (Dynamiker Biotechnology). Among these samples, the β-1-3-D-glucan (BDG) assay (Fungitell), a laboratory standard for the diagnosis of invasive candidiasis, was positive in 20 (40%), intermediate in five (10%) and negative in 25 (50%). In cases of proven or probable candidemia, the sensitivity and specificity of the BDG assay was 86% and 80%, respectively; the *Candida* mannan assay, 14% and 86%; the CAGTA test, 57% and 60%; the *C. albicans* IgM assay, 71% and 60%; and *C. albicans* IgG assay 29% and 90%. In 4/8 (50%) cases with multiple serum samples, *C. albicans* IgM was positive sooner than BDG. Thus, when used as a rule-out test for invasive candidiasis, our data suggest that the *C. albicans* IgM assay may assist antifungal stewardship (over serum BDG).

## 1. Introduction

Invasive candidiasis, including candidemia, encompasses the most serious manifestations of *Candida* infection. It is the most common invasive fungal disease globally [1]. Candidemia is particularly common among critically ill patients receiving antimicrobials and is associated with a significant mortality risk [2]. The diagnosis of candidemia may be challenging, primarily due to a low yield from blood culture specimens and the inconsistent performance of molecular techniques across different patient populations [3]. Indeed, across multiple cohorts of fungaemic patients, an average sensitivity of 38% of has been reported in this setting [4]. Several species-specific and non-specific serological assays are commercially available for the diagnosis of invasive candidiasis: *C. (Candida) albicans* germ tube antibody (CAGTA) assay [renamed as Invasive Candidiasis (CAGTA) IFA]; *C. albicans* IgM (immunoglobulin M); *C. albicans* IgG; and *Candida* mannan ELISA (enzyme-linked immunosorbent assay). These assays are not in routine clinical use. β-1-3-D-glucan (BDG), a carbohydrate biomarker expressed by many pathogenic fungal species, including *Candida*, is used widely in clinical practice to exclude a diagnosis of invasive candidiasis [5]. The primary aim of this study was to compare the diagnostic value of serum BDG versus other biomarker and serological tests for the diagnosis of candidemia.

## 2. Materials and Methods

Serum samples (*n* = 866), taken between March 2013 and July 2016, for BDG assay were identified retrospectively from the database of the Mycology Reference Centre Manchester (MRCM), Wythenshawe Hospital, Manchester, UK. All samples with a minimum of 500 µL excess serum (stored at −70 °C within four hours of venepuncture—in accordance with the manufacturers’ instructions for all assays) from patients with matching blood culture results and with at least two BDG results were included. Cases of invasive candidiasis were classified as “proven” (based on a positive culture for *Candida* sp. from blood or another sterile site) or “probable” (based on the presence of risk factors for the development of invasive candidiasis, a compatible clinical syndrome and treated as such but no positive culture)—based on international consensus definitions [6,7]. Additionally, sera from ten patients tested for BDG but for whom the clinical suspicion of a deep-seated *Candida* infection was very low, and whose blood cultures were negative for *Candida* spp. were included as controls. For proven and probable cases, day zero was defined as the day of sampling of the first positive culture result, or the onset of clinical suspicion, respectively; for controls, this was defined as the timing of the first BDG sample collection.

Four proprietary assays, certified for invasive candidiasis in vitro diagnosis, were compared with serum BDG. Serum BDG was measured using a protease zymogen-based, colorimetric assay (Fungitell^®^, Associates of Cape Cod, East Falmouth, Falmouth, MA, USA). BDG values of <60 pg/mL were defined as negative; 60–79 pg/mL were defined as intermediate; and ≥80 pg/mL as positive. The presence of CAGTA was detected using a specific IgG IFA (immunofluorescence assay) (Virclia^®^, Vircell S.L., Granada, Spain) with a cut-off titre of ≥1:160 for positives. *Candida* mannan was measured using a competitive plate ELISA [Dynamiker^®^, Dynamiker Biotechnology (Tianjin) Co., Ltd., Tianjin, China] using positive and negative cut-off values of 100 and 50 pg/mL, respectively. Readings falling between these values were deemed to be indeterminate and non-diagnostic. *C. albicans*-specific IgM and IgG was measured using separate indirect ELISAs [Dynamiker^®^, Dynamiker Biotechnology (Tianjin) Co., Ltd., Tianjin, China] using positive and negative cut-off values of 120 and 80 AU (antibody concentration)/mL, respectively. Readings falling between these values were taken as intermediate. All assays were used in accordance with the manufacturer’s instructions.

Performance characteristics [sensitivity, specificity, positive predictive value (PPV) and negative predictive value (NPV)] of each assay were calculated. Figures showing BDG and ELISA antigen/antibody levels were plotted using GraphPad Prism^®^ 7 (GraphPad Software, Inc., Avenida De La Playa, CA, USA). Due to the number of patients involved, proven and probable invasive candidiasis groups were merged for data analysis. An indeterminate/intermediate result was regarded as negative for the purpose of sensitivity and specificity analysis.

## 3. Results

The final analysis included 50 serum samples from 24 patients. Of these, 14 had evidence of invasive candidiasis (proven or probable cases). Of these, 20 (40%) were positive for BDG, 5 (10%) were indeterminate and 25 (50%) were negative. BDG was negative in 2 out of the 14 (14%) proven or probable cases and 8 out of the 10 (80%) cases without evidence of invasive *Candida* infection (controls).

The performance of the five assays is summarised in Figure 1. The sensitivity and specificity of BDG for invasive candidiasis in the study population was 86% and 80%, respectively. For the *Candida*-specific assays, sensitivity and specificity were as follows: CAGTA, 57% and 60%, respectively; *Candida* mannan, 14% and 80%, respectively; *Candida* IgM, 71% and 60%, respectively; and *Candida* IgG, 29% and 90%, respectively.

The temporal relationship between BDG, *Candida* IgM/IgG levels and the timing of the first positive deep culture result are outlined in Figure 2. Six proven (A–E) and two probable (F–G) cases (with at least two test results available for analysis) are presented. Time zero represented the date when the first positive sterile site sample was reported. Of the 14 proven or probable cases, 12 had one or more positive biomarker or serological test results before the first sterile site culture sample was collected, which later became positive. Among these, BDG was positive in six (50%), *Candida* IgM in four (33%), CAGTA in three (25%) and *Candida* IgG in one (8%). *Candida* mannan was negative in all such cases. We modelled assay performance against an assumed invasive candidiasis incidence rate of five cases/100,000 bed days (with a mean length-of-stay (LOS) of seven days)—a metric quoted previously in data from our unit [8]. The positive PPV and NPV for BDG were 0.15% and 99.9%, respectively; and 0.062% and 99.8%, respectively, for *Candida* IgM.

## 4. Discussion

Blood cultures remain the gold-standard method for diagnosing *Candida* bloodstream infections, whilst the direct culture of specimens from other sterile sites (such as joint or cerebrospinal fluid) is preferred for non-fungaemic patients [4]. However, the poor real-world performance of direct culture methods—namely, a low specificity and protracted turn-around times—limits their utility to diagnose clinical infection and, therefore, to direct antifungal stewardship (AFS) interventions [9]. BDG is a sensitive fungal biomarker, validated across several patient populations, and currently used as a component of the diagnostic criteria for invasive fungal diseases, including candidemia [6,7,10]. However, BDG is a non-specific biomarker, and false positives due to antigenemia in the absence of an invasive mycosis are relatively common [11]. Moreover, many fungal species express BDG, reducing the utility of the assay for the specific diagnosis of invasive candidiasis [12,13]. Indeed, our data confirm the non-specific nature of BDG for the diagnosis of invasive candidiasis. *Candida*-specific tests (including PCR (polymerase chain reaction) and T2 MR (magnetic resonance) assays) are available. However, due to the low concentration of fungal cells per mL of blood during candidemia, their sensitivity is moderate at best. They also lack sufficient validation and standardisation data to support their use in routine clinical practice [14,15]. Whilst such novel assays may offer advantages by reducing inappropriate anti-fungal use, significant up-front costs and the need for specialist technical and interpretative expertise are likely to preclude their use in non-specialist settings and developing countries for the foreseeable future. Thus, there is a pressing need for specific and cost-effective assays for the diagnosis of invasive candidiasis.

Whilst the NPV of BDG (99.9%) and *Candida* IgM (99.8%) is comparable and within a range acceptable for use in a clinical setting, our data demonstrate that *Candida* IgM is detectable earlier in the course of infection compared to other biomarkers (Figure 2). However, this may be confounded by other factors, including IgM cross-reactivity, which may account for the low specificity of this test when used in this setting. Whilst the CAGTA assay has been suggested as an effective diagnostic marker for invasive candidiasis [16,17], its performance across different populations is questionable [18]. Also, it is specific to *C. albicans,* whilst candidemia due to other species is relatively frequent. However, other data suggest that the CAGTA assay may be more reliable when used as an adjunct to either BDG or mannan testing and may add diagnostic value to either assay [17]. Nevertheless, a review of observational studies highlights the importance of combining mannan antigen and anti-mannan antibody to optimise diagnostic performance, possibly a strategy that prevents assay underestimation of serum antibody-bound mannan versus free mannan [19]. Similar data have been reported regarding combining BDG with *Candida* PCR in patients with invasive candidiasis [15,20]. Thus, a single-assay approach may not represent the optimal strategy.

Regarding biomarker and antibody kinetics during invasive infection, our data suggest that mannan antigen and *Candida* IgG are not readily detectable during the earliest stages of invasive candidiasis. Whilst this is in keeping with the normal immunological response in respect of IgG, which is typically delayed following initial antigenic exposure, it is noteworthy that mannan antigen detection is even less sensitive than *Candida* IgG, despite mannan being one of the main components of the *Candida* cell wall [21,22]. Thus, it would seem intuitive that mannan should be abundant during the earliest stages of bloodstream infections. However, given the clear presence of anti-mannan IgM, it is possible that mannan agglutination by specific IgM isoforms, with or without the formation of immune complexes, may confound early antigenic detection by obscuring epitope–antibody binding within the assay—as has been demonstrated in earlier data from studies involving cases of human disease [23] and in a murine model [24]. Moreover, mannan was found to bind to plasma albumin in the latter study, leading to sequestration. Thus, both phenomena would explain the low sensitivity of mannan detection in early fungaemia, putatively as low mannan levels are sequestered initially, becoming detectable by such techniques only when saturation of physiological sites occurs as the infection progresses and the fungaemic load increases. It would be useful to assess whether such phenomena are indeed important in the laboratory detection of mannan, through assays with extraction or dissociation steps to detect free antigen. This may increase the sensitivity of mannan detection early in invasive infection, thus increasing the value of mannan as a rule-out test. However, assessing the impact of early antifungal therapy, which may reduce the burden of circulating free (unbound) antigen, would be imperative for the diagnostic value of such assays.

We acknowledge limitations to this study. Firstly, this work is limited by the relatively small sample size, especially for proven and probable cases. This is a function of the low sensitivity of blood culture in diagnosing candidemia and our institution’s relatively low incidence-rate [4,8]. Furthermore, sampling was not conducted at fixed time points in the course of infection; thus, a cogent assessment of biomarker kinetics during invasive candidiasis cannot be drawn from our data. However, this study represents a pilot study to observe the quantitative changes in *Candida* biomarkers and antibodies during antifungal treatment. Thus, further studies should explore the serum samples from patients with proven IC and be employed in future studies to develop a complete picture of the changes in *Candida*-related biomarker levels. We also acknowledge the lack of a patient-specific dataset, a necessity given the retrospective and multi-site nature of the samples received in our unit.

## Figures and Tables

**Figure 1 jof-09-00813-f001:**
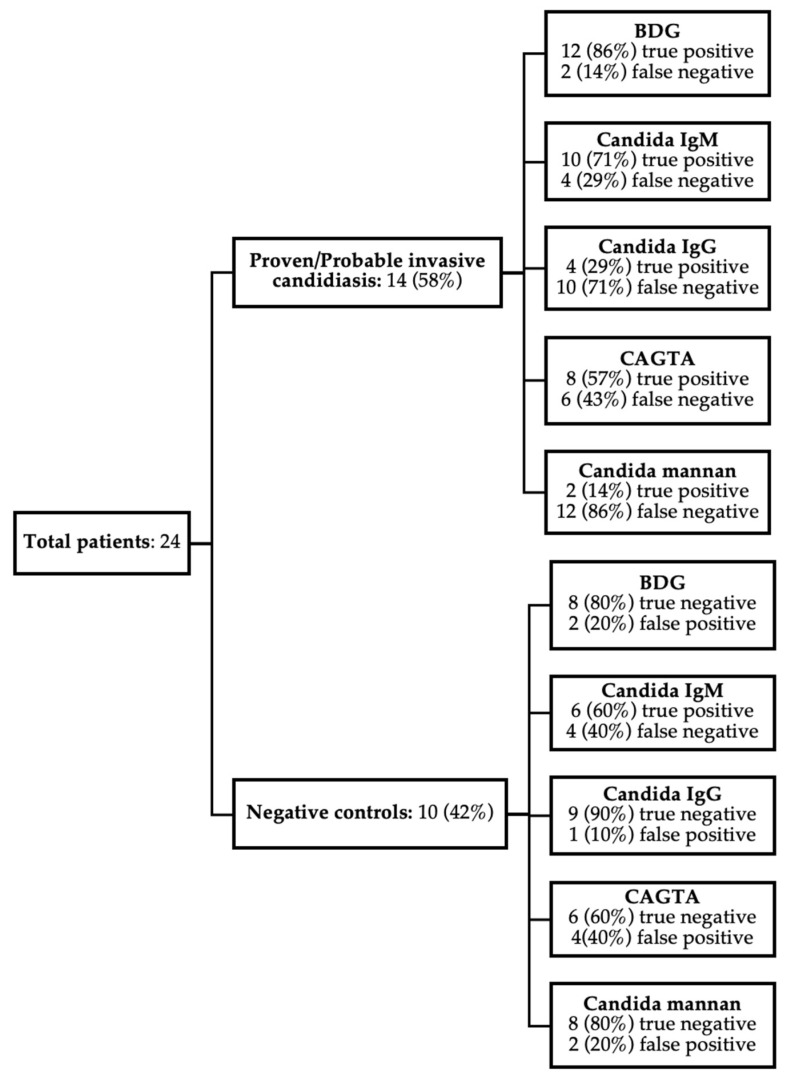
Summary of BDG, *Candida* IgG + IgM, CAGTA and mannan assay results. True/false positive negative rates for 24 patients were shown. Proven and probable cases were grouped together for data analysis.

**Figure 2 jof-09-00813-f002:**
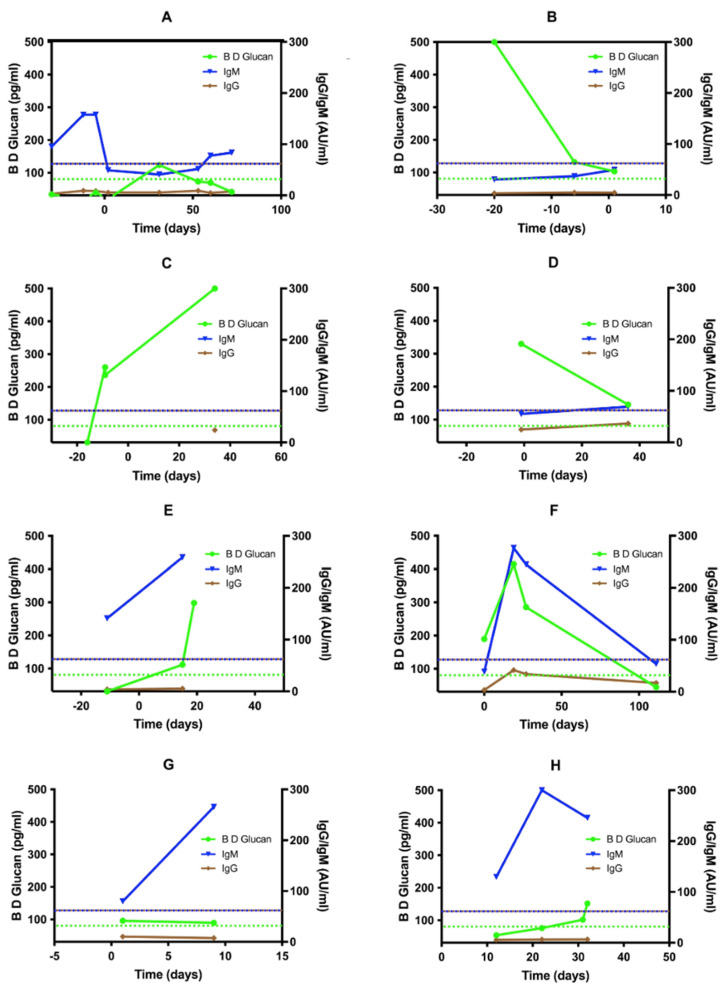
BDG, *Candida* IgM and IgG levels relative to the time of first positive deep culture result. Six proven (**A**–**F**) and two probable (**G**,**H**) cases with multiple sample results are presented. Time zero was defined as the date when positive blood culture was first reported.

## Data Availability

A published dataset is not available due to patient confidentiality concerns.

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
