# Peer review of "Comparison of β-1-3-D-Glucan and Candida Mannan Biomarker Assays with Serological Tests for the Diagnosis of Candidemia"

_jof, 2023, doi:10.3390/jof9080813_

Round 1

Reviewer 1 Report

Manuscript “Comparison of β-1-3-D-glucan…” by Christopher P Eades et al is a single UK center, retrospective study comparing the performance of five commercial serological assays in the diagnosis of candidaemia. It concludes that there were significant differences in the performances of the five assays. The study is well conducted and written. One limitation of the study is the small sample of the proven/probable candidaemia cases, but this is a regular feature in similar studies due to the fact that such cases are rare (the Authors are also recognizing this limitation in the manuscript). Another limitation, due to matters of confidentiality, is the lack of a patients’ data set.

One understands that BDG was performed at the time of the diagnosis and the rest of the assays later. Please clarify when the tests were performed. If the latter is true, please comment on the stability of the biomarkers in the frozen sera.

Minor mistakes

Line 46, “a diagnose invasive”; please correct

Please italicize all scientific names of organisms such as “Candida” or “C. albicans”

Author Response

We are most grateful for the helpful comments of Reviewer 1. We have addressed the points raised by the reviewer as follows:

  1. We acknowledge the reviewer's insightful comments regarding the limitation of the study regarding the lack of a specified data set of patient characteristics. The manuscript has been amended to indicate this explicitly (lines 205-6).
  2. We are grateful for the reviewer's observations regarding the stability of the non-BDG assays in frozen sera. The manuscript has been amended to clarify that the manufacturers' instructions allow for this specifically (lines 57-58). 
  3. As suggested, all scientific names of organisms have been italicised
  4. Line 46 has been amended to redress this typo. We are grateful for this.

Reviewer 2 Report

Some limitations: the sample is very small, there is no accurate diagnosis, low sensitivity for the test. 

The authors would have to do more experiments. Their results are in 10 patients and 10 control. They say in the conclusions that it is a pilot study. They also mention that more studies need to be done. They must make major corrections. Can they increase the number of samples? In that case, wait for new data.

Author Response

We are grateful for the comments of Reviewer 2. As the reviewer quite correctly states, our study is intended to be a pilot to give the readers of the Journal of Fungi an overview in respect of the use of these novel assays in the diagnosis of invasive candidiasis as compared to a common diagnostic standard (i.e., beta-D-glucan). As stated in the manuscript and reflecting existing, peer-reviewed data, invasive candidiasis is rare in our centre, indicative of the UK epidemiology. For this reason, and in keeping with similar published data, we feel the sample size is entirely in keeping with what one would expect from a preliminary evaluative study. 

In respect of the observations of the reviewer apropos the "low sensitivity of the test". We are grateful for this observation but can only reassert the commentary within the manuscript (specifically, lines 155-170), in which the poor performance of specific assays in this context is evaluated. Indeed, such observations are in keeping with other published data, as referenced in the manuscript. 

In summary, whilst the limitations of the work are clearly acknowledged within the text, we believe our manuscript meets the standards for publication in the Journal of Fungi. We are grateful for the insightful and informative comments of the reviewer in this regard. 

Reviewer 3 Report

As the selected reference biomarker (β-1-3-D-glucan) is non-specific, the need to specify its use for candidaemia is misleading. Perhaps the study would be better conducted if it focused on the diagnosis of yeast infections in the blood in general, adding analyzes with other markers from other fungal species.

Author Response

We are very grateful to the helpful comments of Reviewer 3 in respect of our manuscript. As referenced in the manuscript, Beta-D-glucan (BDG) is a common biomarker used as a "rule-out" test for invasive candidiasis, including bloodstream infections. Indeed, within the manuscript, we reference the use of the biomarker as a trigger for stopping antifungals in high-risk populations, including critical care. In our centre, and reflecting wider clinical practice within the United Kingdom, BDG is used clinically as a screening test for invasive candidiasis - as referenced in the manuscript 

Whilst we acknowledge the non-specific nature of BDG for this indication (lines 141-144), in the context of the broader UK epidemiology (i.e., where non-Candida bloodstream infections are exceedingly uncommon), we re-affirm the use of the biomarker as a comparator for the other assays. Moreover, we have stated explicitly that other Candida-specific assays (including T2MR and Candida PCR) are not necessarily superior to BDG due to poor sensitivity (lines 143-148).

We are most grateful for the insightful comments, but taken together, we feel confident in re-affirming our view that the manuscript is suitable for publication in the Journal of Fungi in its current format. 

Round 2

Reviewer 2 Report

Aceppt in this form

Author Response

We are grateful for the comments of the reviewer and the final decision to accept.